# Application of ^18^F-FDG Positron Emission Tomography/Magnetic Resonance in Evaluation of Oropharyngeal Carcinoma

**DOI:** 10.3390/diagnostics15091081

**Published:** 2025-04-24

**Authors:** Yilin Shen, Jichang Wu, Chenling Shen, Xinyun Huang, Cui Fan, Haixia Hu, Zenghui Cheng, Biao Li, Mingliang Xiang, Bin Ye

**Affiliations:** 1Department of Otolaryngology & Head and Neck Surgery, Ruijin Hospital, Shanghai Jiao Tong University School of Medicine, Shanghai 200070, China; syl13026@rjh.com.cn (Y.S.); benoit_wu@163.com (J.W.); shen3023824@163.com (C.S.); fc12456@rjh.com.cn (C.F.); hhx12195@rjh.com.cn (H.H.);; 2Department of Nuclear Medicine, Ruijin Hospital, Shanghai Jiao Tong University School of Medicine, Shanghai 200025, China; 3Department of Radiology, Ruijin Hospital, Shanghai Jiao Tong University School of Medicine, Shanghai 200025, China; czh12048@rjh.com.cn

**Keywords:** oropharyngeal carcinoma, tonsil carcinoma, FDG PET/MR, cervical lymph node metastases

## Abstract

**Objectives:** Oropharyngeal carcinoma is experiencing an increase in incidence and can easily metastasize to the cervical lymph nodes. Therefore, evaluating the tumor boundary and lymph node metastasis before treatment is critical. Both CT and MR may have limitations in describing the specific boundaries of oropharyngeal tumors. To date, no research has applied PET/MR imaging to patients with only oropharyngeal carcinoma and verified its diagnostic value. The aim of our study was to evaluate the diagnostic value of PET/MR in patients with oropharyngeal carcinoma. We prepared PET/MR for comparison with CT/MR for T and N staging, with the aim of exploring the relationship between the imaging parameters and different biological factors. **Methods:** This was a retrospective, observational study. In total, 13 patients (11 males and 2 females) with oropharyngeal tumors who underwent FDG PET/MR and enhanced CT/MR from July 2021 to December 2022 were retrospectively analyzed. Cohen’s kappa coefficient and the McNemar test were used to compare the consistencies and diagnostic values of FDG PET/MR and enhanced CT/MR imaging in relation to primary tumors and cervical lymph node metastases. Various specific parameters of FDG PET/MR were included in the statistics. Spearman correlation coefficients were used to analyze the relationship between the parameters and the tumor stage, the degree of differentiation, p16 expression, Ki67 expression, and serological tumor markers. **Results**: The average age of the patients was 61.54 ± 6.62 years old. Preoperative imaging demonstrated good consistency between FDG PET/MR and enhanced CT and MR for the diagnosis of clinical T stage. A total of seven patients underwent surgery directly. Overall, 231 cervical lymph nodes were dissected. Compared to the postoperative histopathological results, PET/MR was significantly more sensitive than enhanced CT/MR imaging (78.57% vs. 50.00%, *p* < 0.05; 78.57% vs. 64.29%, *p* < 0.05, respectively). Also, PET/MR showed more accuracy in diagnosing metastatic lymph nodes, but without significance. Combined with PET/MR-specific parameters, the SUV, TLG, and the MTV were found to be higher in the patients with more advanced stages of cancer and lower in those with p16-positive tumors. In addition, they were found to be positively correlated with the level of serum CEA. **Conclusions:** This is the first study to evaluate the clinical diagnostic value of PET/MR in patients with oropharyngeal carcinoma. We believe that PET/MR has more advantages in describing tumor boundaries. It is more sensitive or even more accurate for the evaluation of metastatic cervical lymph nodes.

## 1. Introduction

The incidence of oropharyngeal carcinoma is about 8–10/100,000, and squamous cell carcinoma of the oropharynx accounts for approximately 25% of all head and neck squamous cell carcinomas. Tobacco and alcohol consumption and HPV infection are the main risk factors [1]. Oropharyngeal carcinoma can easily metastasize to the cervical lymph nodes. Thus, the evaluation of the tumor boundary and lymph node metastasis before treatment is critical. The treatments for oropharyngeal carcinoma include surgical treatment, radiotherapy, and a combined treatment. TN staging according to the guidelines of the AJCC in 2017 is an important tool that guides clinicians to making a therapeutic strategy [2]. Thus, high-quality preoperative imaging is necessary to provide precise TN staging. A recent study showed that MR imaging was indispensable for the accurate localization and description of the surrounding extension of the oral cavity and oropharyngeal tumors [3]. Both MR and PET/CT are recommended in the latest 2022 NCCN Head and Neck Cancer guidelines [4]. However, although PET/CT provides superior metabolic characterization, it has not shown a difference compared to MR in the local staging of oropharyngeal cancer [5]. In order to better display soft tissue and reserve the metabolic sensitivity of PET, the first PET/MR scanner was installed for clinical use in 2010 [6]. The radiation dose of PET/MR was reduced by about 73% compared with that of PET/CT [7], suggesting that PET/MR has a better application prospect. It has been applied to more than 16 kinds of malignant tumor, and its diagnostic efficacy has been proved to be better than or at least similar to that of PET/CT in evaluating incidental masses of the abdominal organs [8]. Reportedly, PET/MR has been applied to head and neck tumors, including nasopharyngeal carcinoma, oropharyngeal carcinoma, and laryngeal and pharyngeal carcinoma at the same time [9]. However, great differences exist in the pathological characteristics and the biological behaviors among these tumors, and the TN staging systems are not the same, so it may be inaccurate to integrate them into one study. This study intends to retrospectively analyze whether PET/MR shows advantages in the preoperative diagnosis of oropharyngeal malignant tumors compared with enhanced CT and MR. Furthermore, various imaging biomarker measurements of PET/MR, including the standard uptake value (SUV), TLG, the MTV, and the ADC, are analyzed to explore their association with p16, the tumor differentiation degree, and the serologic markers.

## 2. Materials and Methods

### 2.1. Patients

This study was approved by the institutional review board of Ruijin Hospital affiliated with Shanghai Jiaotong University (KY2023 LS No. 067). Our study consisted of 13 patients with biopsy-proven oropharyngeal carcinoma who underwent both enhanced CT/MR and PET/MR between July 2021 and December 2022. Patients with T1–T4 stages were all included, along with patients who were both HPV-positive and HPV-negative. The tumor locations included the tonsil (*n* = 10), the tongue (*n* = 2), and the posterior wall of the oropharynx (*n* = 1). Patients were excluded if they had a prior history of radiation or chemotherapy.

### 2.2. Imaging

**PET/MR** The protocol is as same as that in the previous studies by our research team from the nuclear medicine department in Ruijin Hospital [10]. Briefly, all the patients underwent FDG PET/MR with a hybrid PET-MR Siemens Biograph mMR scanner (Siemens Healthineers, Erlangen, Germany). The patients were required to fast for more than 6 h before imaging. They received an IV injection of 18F−FDG at a mean dose of (184.56 ± 41.19) MBq [10]. Simultaneous FDG PET/MR acquisition started 40 min after injection. The patients were asked to avoid talking during the uptake time of the radiopharmaceutical to avoid impacting the evaluation of the primary tumor. During the scan, the patients were asked to remain calm and minimize any movement.

**CT and MR** Diagnostic MR was performed using a 1.5 T scanner (Avanto, Siemens AG, Healthcare Sector, Erlangen, Germany). The MR protocol used is as follows: (1) A nonenhanced T1 turbo spin echo (TSE) sequence in coronal orientation (TR 558 ms, TE 10 ms, slice thickness 5 mm, FOV 300 mm). (2) A nonenhanced T1 TSE sequence in transverse orientation (TR 544 ms, TE 11 ms, slice thickness 3 mm, FOV 260 mm). (3) A T2 TSE sequence in transverse orientation (TR 4, 2000 ms, TE 125 ms, slice thickness 3 mm, FOV 260 mm). (4) A contrast-enhanced T1 TSE sequence in coronal orientation (TR 669 ms, TE 10 ms, slice thickness 5 mm, FOV 300 mm). (5) A DWI sequence in coronal orientation (TR 5000 ms, TE 53 ms, slice thickness 5 mm, FOV 380 mm). CT scans were obtained in the axial plane using various models of with 5 mm section thickness. Images were obtained both before and after the intravenous administration of iodinated contrast material (gadodiamide; Shanghai Kexing Biopharm, Shanghai, China).

### 2.3. Imaging Analysis

The FDG PET/MR results were interpreted by a board-certified nuclear medicine physician (with 5 years’ experience in head and neck radiology) using Integrated Registration software in syngo.via VB2.0 Workstation. The image interpretation included regional analysis and quantitative assessment. According to the location of the tumors, the lesions were divided into primary tumors and lymph node metastasis. Two nuclear medicine physicians evaluated the image qualities according to the presence of geometric distortion, swallowing, or breathing artifacts. The focal uptake area was assessed in terms of alignment between the PET and anatomic structures, the quality of fusion, lesion conspicuity, and anatomic location. Lesions with focal uptake were defined as well-circumscribed areas of increased tracer uptake relative to the surrounding structures. The physicians classified the areas of focal uptake as probably malignant or benign based on visual analysis, the asymmetry of tracer uptake, and consideration of the normal distribution of 18F-FDG in the head and neck region. The MR and CT images were, respectively, analyzed by two radiologists specialized in head and neck imaging and were blinded to the experimental groups. If the radiologists disagreed, a third radiologist with >30 years of experience was consulted and provided the final decision. For the purposes of this study, the relevant imaging biomarker measurements were the SUV (max), the SUV (mean), the SUV (peak), the ADC (min), the ADC (max), the ADC (mean), the MTV, and TLG from PET/MR of all the primary oropharyngeal lesions and metastatic lymph nodes. For the ADC measurements (the ADC (mean), the ADC (max), and the ADC (min)), the regions of interest (ROIs) were manually drawn on ADC maps along the tumor or lymph node contour on a single slice from the largest area of the tumor or lymph node. The ROIs were 6.1 ± 3.0 cm^2^ (range: 1.2–14.7 cm^2^). The relevant PET-related parameters measured included the SUV (max), which reflected the maximal SUV (adjusted for body weight); the SUV (mean), which represented the mean SUV within the ROIs; and the SUV (peak), which referred to the computationally automated maximal average SUV in a 1 cm^3^ spherical volume within a tumor [11]. The MTV was defined as the tumor volume with 18F-FDG uptake segmented by the PET edge method. TLG was defined as (MTV) × (SUVmean) [12]. The criteria for lymph node metastasis are as follows: (1) the short-axis diameter of lymph nodes in the CT or MR images is greater than 10 mm; (2) the lymph nodes are spherical rather than flat; (3) signs of central necrosis; (4) the abnormal fusion of several lymph nodes; (5) the lymph nodes show a strong signal on DWI and a low value of the ADC in the MR images; (6) for visual analysis, the lymph node is considered abnormal if 18F-FDG uptake is greater than the background activity and is considered malignant when the SUV (max) is greater than 2.5 [13,14,15].

### 2.4. Surgical Procedure and Histopathology

All the patients underwent the biopsy of the oropharyngeal tumor and the detection of p16 before further treatments. Seven patients (all T1 or T2) directly received surgery and neck dissection. Three patients received both chemotherapy and radiotherapy effectively to reduce tumor mass before a surgical option was presented. Three patients only received radical chemotherapy as palliative care. The extent of neck dissection was determined according to the tumor location, stage, and the preoperative imaging findings. The patients with no nodal involvement underwent functional neck dissection at levels I–III or II–IV depending on the location of the tumor. The patients with one or more involved lymph nodes or extranodal extension underwent radical neck dissection at levels I–V [16]. The histologic grade of tumors was determined by a pathologist (with over 10 years’ experience in pathology) without knowing any CT, MR, or PET/MR results. Additionally, for primary tumor lesions, p16 expression was detected as a surrogate marker for HPV infection using immunohistochemical skills. Also, ki67 protein was detected to indicate proliferation. The HPV subtype was examined using the PCR method.

### 2.5. Statistical Analysis

All data analyses were performed using SPSS Statistics (version 21.0). Two-tailed P values less than 0.05 were considered statistically significant. For primary tumors and neck lymph nodes, the diagnostic accuracy of T staging and N staging calculated for PET/MR, enhanced CT, and MR alone is presented according to the reference standard. Differences among the 3 modalities were compared using the McNemar test. The sensitivity, specificity, positive predictive value (PPV), negative predictive value (NPV), and accuracy of each imaging modality (PET/MR, enhanced CT, and MR) were calculated using the histopathology results from the surgical specimens as a gold standard reference. Cohen’s kappa was used to evaluate inter-rater agreement. A value from 0 to 0.19 was defined as slight, from 0.2 to 0.39 as fair, from 0.4 to 0.59 as moderate, from 0.6 to 0.79 as good, and from 0.80 to 1 as almost excellent agreement [17]. The ADC, the SUV, TLG, and the MTV of the primary tumor lesions for PET/MR and p16, Ki67, and the tumor markers were compared using Spearman correlation coefficients. Correlation coefficients higher than 0.7 were regarded as strong, and between 0.3 and 0.7 as moderate. A value of 0.3 or less represented a weak correlation [18].

## 3. Results

### 3.1. Clinical Characteristics of Patients with Oropharyngeal Cancer

The patient characteristics and their clinical features are shown in Table 1. A total of 13 patients were included. There were 11 male (84.6%) and 2 female (15.4%) patients, and the median age was 61.54 ± 6.62 years. The most common site for primary tumors was the tonsil (*n* = 10, 76.9%), followed by the tongue (*n* = 2, 15.4%) and the posterior wall of the oropharynx (*n* = 1, 7.7%). The only histologic type was squamous cell carcinoma. Among the thirteen patients, seven patients (53.8%) showed p16 (+), and six patients (46.2%) showed p16 (−). For tumor differentiation, four patients (30.7%) showed poor differentiation, six patients (46.2%) showed moderate differentiation, and three patients (23.1%) showed strong differentiation. Seven patients (53.8%) had pathologic stage T1 or T2 lesions and underwent surgery directly. Three patients (23.1%) received neoadjuvant chemoradiotherapy before surgery, and another three patients (23.1%) received radical radiotherapy. Among the seven patients with the pathologic stage who underwent surgery directly, ipsilateral cervical metastases were present in four (57.1%), bilateral involvement was present in two (28.6%), and no cervical metastasis was present in one (14.3%). No distant metastasis was found through PET/MR.

### 3.2. Comparison of Clinical TN Staging Among Enhanced CT, MR and PET/MR Imaging Before Treatment

T and N staging was performed based on the 8th edition of the AJCC TNM staging system. The tumors in the 13 patients were all confirmed to be malignant by PET/MR according to the criteria for lymph node metastasis [14,15,16]. Firstly, the diagnostic performances of enhanced CT, MR, and PET/MR before surgery were compared. The preoperative TN staging of oropharyngeal carcinoma was exhibited (Appendix A). For T staging, Cohen’s κ demonstrated perfect inter-reader agreement between PET/MR and enhanced CT (κ = 1, *p* < 0.000), as well as good inter-reader agreement between PET/MR and enhanced MR (κ = 0.87, *p* < 0.000). For N staging, Cohen’s κ demonstrated moderate inter-reader agreement between PET/MR and enhanced CT (κ = 0.567, *p* = 0.001), as well as moderate inter-reader agreement between PET/MR and enhanced MR (κ = 0.577, *p* < 0.000) (Table 2). Since T staging in oropharyngeal tumors is based on tumor size, preoperative imaging is extremely important for the evaluation of tumor size. The maximum transverse diameter of the tumor was then measured by enhanced CT, MR, and PET/MR (Appendix A). The mean diameters were 2.76 cm, 2.84 cm, and 2.69 cm, respectively, with no significant difference (*p* = 0.907). However, we found that the estimated tumor size on enhanced CT or MR was usually larger than that on PET/MR (Figure 1 and Figure 2).

### 3.3. Comparison of Diagnostic Accuracy of Cervical Lymph Nodes Among Enhanced CT, MR and PET/MR Imaging

Since most patients who underwent surgery directly were at T1 or T2 stage, we further compared the accuracies of the diagnosis of metastatic cervical lymph nodes in the patients receiving surgeries. The image data were interpreted for the evaluation of cervical lymph node involvement, and then were compared to the pathological results. Overall, 231 sides were dissected, with 160 (69.3%) and 71 (30.7%) sides dissected in ipsilateral and contralateral necks, respectively. In total, the pathological results showed 14 positive lymph nodes and 217 negative lymph nodes after the surgeries. The findings of PET/MR, enhanced CT, and MR are shown in Appendix A. PET/MR was significantly more sensitive than enhanced CT/MR imaging (78.57% vs. 50.00%, *p* < 0.05; 78.57% vs. 64.29%, *p* < 0.05, respectively). Also, PET/MR was more accurate than enhanced CT/MR imaging, but without significance (97.65% vs. 94.84%, *p* > 0.05; 97.65% vs. 96.71%, *p* > 0.05, respectively). More cases needed to be selected in the future. However, PET/MR showed no advantages in the specificity of diagnosis (Table 3).

### 3.4. Relationship Between ADC/SUV/TLG/MTV and Ki67 as Well as Serological Markers

Ki67 is a common marker for the evaluation of cell proliferation and is correlated with the degree of tumor malignancy. Up till now, no serological tumor marker has been found to be specific to oropharyngeal carcinoma, so common biomarkers, including alpha-fetoprotein (AFP), carcinoembryonic antigen (CEA), neuron-specific enolase (NSE), cytokeratin 19, squamous cell carcinoma-associated antigen (SCC), CA125, CA724, CA199, and CA242, were examined in the blood samples of the patients at least 48 h before receiving PET/MR examination. Abnormal tumor markers, such as CEA, NSE, cytokeratin 19, and SCC, were chosen for further research. The relationship between the PET/MR parameters (the ADC, the SUV, TLG, and the MTV) and Ki67 or serological markers were explored using the Spearman correlation coefficient. A significant negative correlation was found between the ADC (max) and Ki67 (r = −0.652, *p* = 0.041). The ADC (mean) value was found positively correlated with NSE (r = 0.794, *p* = 0.019). The SUV (peak), TLG, and the MTV all showed a positive correlation with CEA (r = 0.803, 0.895, 0.762, *p* = 0.016, 0.003, 0.028) (Table 4).

### 3.5. Relationship Between ADC/SUV/TLG/MTV and Pathological TN Stage

In order to explore the relationship between various PET/MR parameters and the pathological TN staging of tumors, the ADC (min), the ADC (max), the ADC (mean), the SUV (max), the SUV (peak), the SUV (mean), TLG, and the MTV of the primary tumor and the metastatic cervical lymph nodes were analyzed by the radiologists using the *t*-test or one-way ANOVA. No significant difference was found. However, we observed a trend; compared with the tumors of T1 stage, the tumors of T2 stage showed a higher SUV, TLG, and the MTV. In the different N stages, no significant changing trend was found (Appendix A).

### 3.6. Relationship Between ADC/SUV/TLG/MTV and P16

p16 is an important prognostic marker related to HPV infection in oropharyngeal carcinoma. HPV continuously infects the squamous epithelia, causing abnormal cell proliferation and the over-expression of p16. In recent years, the incidence of HPV-positive tonsillar cancer has shown a rapid rising trend [19]. The therapeutic effect and prognosis are better than those of HPV-negative carcinoma. We further analyzed the relationship between the PET/MR parameters and p16 and found that the p16-positive tumors showed a trend of lower SUV parameters, TLG, and MTV, although their difference was not significant (Appendix A).

### 3.7. Relationship Between ADC/SUV/TLG/MTV and Tumor Differentiation

Tumor differentiation plays an important role in the prediction of prognosis and guidance for the comprehensive treatment of oropharyngeal malignant tumors [20]. The SUV and TLG gradually increased in the moderately and well-differentiated tumors compared with those of the poorly differentiated tumors. The ADC in the well-differentiated group was higher than those in the poorly or moderately differentiated groups. However, there was also no significant difference among the three groups (Appendix A).

## 4. Discussion

The diagnostic efficacy of PET/MR in the different types of head and neck squamous cell carcinoma (HNSCC) varies largely. In a comprehensive study involving the tongue, the larynx, and the hypopharynx, no significant difference was found among PET/MR and PET/CT and MR [21,22]. However, in one specific type of nasopharyngeal cancer, PET/MR was found to be more accurate than PET/CT or MR regarding the N staging assessment [23]. In hypopharyngeal cancer, PET/MR was found to be more accurate in T staging for primary tumors, but did not show significant advantages in N staging [18]. Our research is the first retrospective study to compare the application of PET/MR with enhanced CT/MR in oropharyngeal carcinoma alone. For preoperative TN staging diagnosis, the results showed that PET/MR was highly consistent with those of enhanced CT and MR for T staging. Schaarschmidt et al. also reported a similar diagnostic accuracy among PET/MR, PET/CT, and MR for the primary diagnosis of head and neck carcinoma or the detection of recurrence [21]. The T stage of oropharyngeal carcinoma is mainly dependent on the maximum diameter of the tumor. No significant difference was found among the three modalities in measuring the tumor size, but the tumor diameter obtained by PET/MR was the smallest, followed by enhanced MR and CT. This finding was similar to previous research that concluded fused PET/MR is superior to MR for the measurement of tumor size and is helpful when evaluating the precise extent of bone resection required to remove the tumor, but preserve function [17]. Queiroz et al. reported that PET/MR could also be more helpful for the evaluation of the oral cavity and the oropharynx to reduce artifacts caused by dental implants [24]. Therefore, we believe that PET/MR can provide more precise imaging guidance for the surgical scope. Also, PET/MR outperformed PET/CT in the workup of suspected occult malignancies [25]. For the detection of lymph nodes, PET/MR could detect a higher number of positive lymph nodes compared to those of CT or MR in cholangiocarcinoma [26] and breast cancer [27]. The fusion of PET/CT and MR was found to raise the accuracy of in the dependent detection of cervical lymph node metastases in HNSCC [28]. But Platzek reported that in HNSCC, FDG PET/MR did not significantly improve the accuracy for cervical lymph node metastases in comparison to that of MR [29]. In our study, we analyzed the diagnostic efficacy of metastatic cervical lymph nodes compared to the histopathologic results. PET/MR was found to be more sensitive than enhanced CT/MR, but it did not show significantly improved accuracy. Combined with the previous results, we recommend the use of PET/MR for the evaluation of metastatic cervical lymph nodes in oropharyngeal carcinoma.

Since PET/MR combines various parameters specific to MR, this was the first time we explored the relationship between these parameters (the SUV, the ADC, TLG, and the MTV) and the tumor TN stage, the differentiation degree, and p16. Some studies have shown that the TLG value is closely related to the late stage of oropharyngeal carcinoma [30]. The MTV and TLG were found to be significantly correlated with overall survival [7]. In our research, the tumors of T2 stage showed a higher SUV, TLG, and MTV compared to tumors of T1 stage, although no significance was found. Since p16 is an independent factor that affects the TN stage and prognosis in oropharyngeal carcinoma [31], the patients with p16-positive expression showed a lower SUV, TLG, and MTV. The ADC is a unique parameter of MR, reflecting the rate of diffusion of water molecules within tissue. It depends on the tissue barrier such as the cell membrane and indirectly reflects the cell density of tissues [32]. We found that the ADC showed a better trend in the well-differentiated group, which was similar to the results from patients with cervical cancer. Gong et al. found that the SUV (max) had a higher diagnostic value in lymph node metastasis and FIGO staging, while the ADC (min) had a higher diagnostic value in pathological differentiation [33]. In addition, the ADC (max) was found to be highly negatively correlated with Ki67 expression (r = −0.652, *p* = 0.041). Meta-analysis about the relationships between different imaging parameters and the histopathology of HNSCC showed that the ADC could predict cell count and proliferation activity and was negatively correlated with Ki67 (r = −0.61) [34]. A similar correlation was also found in other tumors, such as thyroid cancer, esophageal cancer, stomach cancer, and malignant melanoma [35]. In oral squamous cell carcinoma, the ADC showed a significant difference between different Ki67 statuses and had potential as a promising prognostic biomarker [36]. This results suggest that the ADC can better reflect the characteristics of tumor cells and the microenvironment.

No serologic tumor markers have been found to be highly specific to HNSCC so far. In this research, CEA, NSE, cytokeratin 19, and SCC were chosen for further study. Adamiak et al. reported that NSE cannot be used to accurately monitor patients with HNSCC [37], while the CEA level was correlated with the clinical stage of tumor and could decline after resection [38]. In patients with oral squamous cell carcinoma, the CEA levels were significantly increased in saliva and local tumor-exfoliated cells [39]. The CEA and SCC levels decreased after therapy in patients with HNSCC, which could be used to assess the treatment response and prognosis [40]. Many studies proved that there was relationship between the SUV of PET/CT and the tumor markers in different malignant tumors. In patients with non-small-cell lung cancer, a decreased SUV (max) and increased CEA levels were associated with EGFR mutations. They could serve as predictors for the EGFR-TKI therapeutic response [41]. Combining the SUV (peak) of the primary tumor and the SUV (max) of the mediastinal lymph node with serum CEA and SCC could predict mediastinal lymph node metastasis in these patients effectively [42]. In our study, we found that the ADC (mean) was positively correlated with the NSE level; the SUV (peak), TLG, and the MTV were positively correlated with the CEA level. Moreover, the patients at the late stage showed a higher SUV, TLG, and MTV. Thus, we believe that combining these parameters with CEA may better evaluate the clinical stage, therapeutic response, or prognosis of patients with oropharyngeal carcinoma in the future. It would be meaningful to further expand the sample size.

As for the limitations in this study, firstly, considering the high cost of PET/MR, only a limited number of patients can afford it. As a result, the number of patients in each stage is insufficient, and only the patients with T1–T2 tumors received surgeries, which may have caused a statistical bias. More patients need to be included to better verify the diagnostic value of PET/MR in oropharyngeal carcinoma. Additionally, aside from enhanced CT/MR, PET/CT should also be compared with PET/MR to explore whether PET/MR has more diagnostic advantages in oropharyngeal carcinoma, especially in evaluating cervical lymph node metastasis.

## Figures and Tables

**Figure 1 diagnostics-15-01081-f001:**
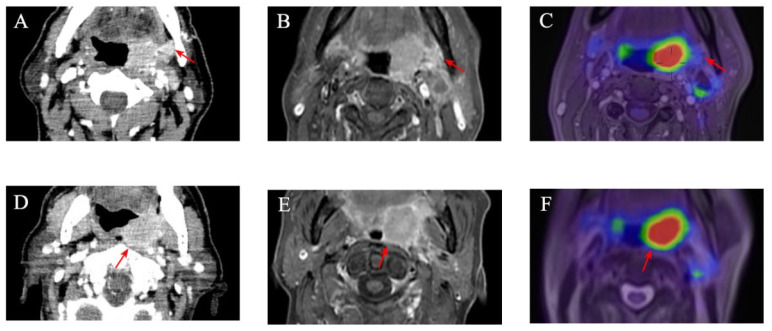
Seventy-year-old male patient with squamous cell carcinoma of left tonsil. (**A**) Contrast-enhanced CT image and (**B**) contrast-enhanced T1-weighted image showing blurred boundary between tumor and mandible (red arrows). (**C**) No infiltration of mandible can be seen on PET/MR image (red arrow). (**D**) Contrast-enhanced CT image and (**E**) contrast-enhanced T1-weighted image showing blurred boundary between tumor and prevertebral space (red arrows). (**F**) No infiltration of prevertebral space by primary can be seen on PET/MR image (red arrow mean tumor boundary).

**Figure 2 diagnostics-15-01081-f002:**
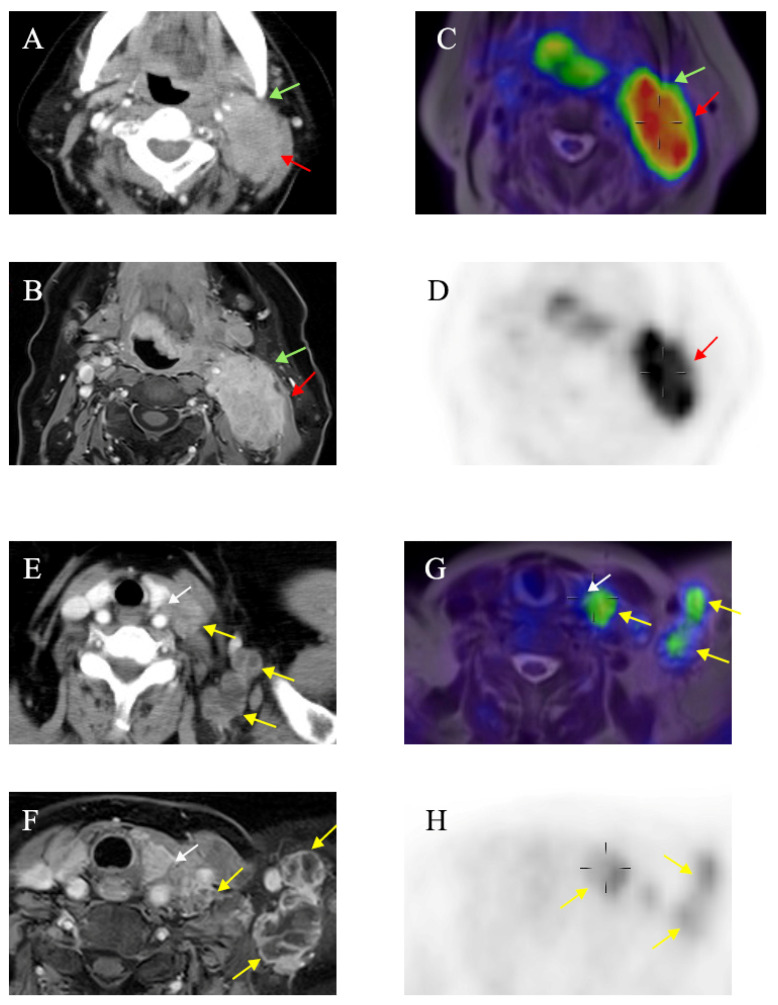
Sixty-five-year-old female patient with squamous cell carcinoma of left tonsil showing extensive cervical lymph node metastasis. (**A**) Contrast-enhanced CT image and (**B**) contrast-enhanced T1-weighted image showing blurred boundary between metastatic lymph nodes and mandible (green arrows). Sternocleidomastoid muscle was also infiltrated by lymph nodes (red arrows). (**C**,**D**) No infiltration of mandible (green arrows) can be seen on PET/MR image (red arrows show boundary between lymph nodes and sternocleidomastoid muscle). (**E**) Contrast-enhanced CT image and (**F**) contrast-enhanced T1-weighted image showing extensive supraclavicular lymph nodes metastasis (yellow arrows). Boundary between metastatic lymph nodes and left thyroid gland is blurred (white arrows). (**G**,**H**) No infiltration of left thyroid gland (white arrows) can be seen on PET/MR image (yellow arrows show extensive lymph nodes metastasis).

**Table 1 diagnostics-15-01081-t001:** Patients’ characteristics.

Characteristic	Data
Sex	
Male	11
Female	2
**Age (y)**	
≤60	7
>60	6
**Primary tumor site**	
Left tonsil	5
Right tonsil	5
Posterior wall of oropharynx	1
Tongue	2
**Histologic type**	
squamous cell carcinoma, HPV (+)	7
squamous cell carcinoma, HPV (−)	6
**Histologic grade**	
poorly differentiated	4
moderately differentiated	6
well differentiated	3
**Treatment**	
Surgery alone	7
Chemoradiation + surgery	3
Radiation	3
**Neck metastasis** (only for patients receiving surgery alone)	
None	1
Ipsilateral	4
Bilateral	2

**Table 2 diagnostics-15-01081-t002:** Consistency analysis between enhanced CT/MR and PET/MR imaging results for clinical TN staging before surgery.

Enhanced CT T Stage	PET/MR T Stage				Total	κ	*p*
	T1	T2	T3	T4			
T1	2	0	0	0	2	1.000	0.000
T2	0	7	0	0	7		
T3	0	0	1	0	1		
T4	0	0	0	3	3		
Total	2	7	1	3	13		
Enhanced CT N stage	PET/MR N stage				Total	κ	*p*
	N0	N1	N2	N3			
N0	1	2	0	0	3	0.567	0.001
N1	0	4	0	0	4		
N2	2	0	3	0	5		
N3	0	0	0	1	1		
Total	3	6	3	1	13		
Enhanced MR T stage	PET/MR T stage				Total	κ	*p*
	T1	T2	T3	T4			
T1	2	0	0	0	2	0.870	0.000
T2	0	7	1	0	8		
T3	0	0	0	0	0		
T4	0	0	0	3	3		
Total	2	7	1	3	13		
Enhanced MR N stage	PET/MR N stage				Total	κ	*p*
	N0	N1	N2	N3			
N0	2	2	0	0	4	0.577	0.000
N1	0	3	0	0	3		
N2	1	1	3	0	5		
N3	0	0	0	1	1		
Total	3	6	3	1	13		

**Table 3 diagnostics-15-01081-t003:** Relationship between enhanced CT/MR imaging and FDG PET/MR results and neck histopathologic results of lymph nodes. PPV: positive predictive value; NPV: negative predictive value.

Analysis	Sensitivity	Specificity	Accuracy	PPV	NPV
Enhanced CT	50.00%	97.99%	94.84%	64.63%	96.53%
Enhanced MR	64.29%	98.99%	96.71%	81.82%	97.52%
PET/MR	78.57%	98.99%	97.65%	84.62%	98.50%

**Table 4 diagnostics-15-01081-t004:** Relationship between PET/MR parameters (ADC, SUV, TLG, and MTV) and Ki67 and serological markers (CEA, NSE, cytokeratin 19, and SCC). “*” means *p* < 0.05 between two groups.

		CEA (ng/mL)	Cytokeratin 19 (ng/mL)	NSE (ng/mL)	SCC (ng/mL)	Ki67
Parameters of tumor						
ADC (min)	r	0.553	0.214	0.590	0.170	−0.029
	*p*	0.155	0.611	0.123	0.688	0.937
ADC (max)	r	0.537	−0.065	0.463	−0.126	−0.652
	*p*	0.170	0.878	0.248	0.766	0.041 *
ADC (mean)	r	0.623	0.164	0.794	0.176	−0.420
	*p*	0.099	0.698	0.019 *	0.677	0.227
SUV (max)	r	0.655	0.446	0.390	0.144	−0.176
	*p*	0.078	0.269	0.340	0.734	0.626
SUV (peak)	r	0.803	0.369	0.395	−0.048	−0.242
	*p*	0.016 *	0.369	0.333	0.910	0.501
SUV (mean)	r	0.666	0.429	0.382	0.126	−0.169
	*p*	0.071	0.289	0.350	0.767	0.642
TLG	r	0.895	0.290	0.377	−0.261	−0.243
	*p*	0.003 *	0.486	0.357	0.532	0.499
MTV	r	0.762	0.216	0.244	−0.399	−0.209
	*p*	0.028 *	0.608	0.561	0.411	0.562

## Data Availability

All data generated or analyzed during this study are included in this published article.

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
