# Peer review of "Application of ^18^F-FDG Positron Emission Tomography/Magnetic Resonance in Evaluation of Oropharyngeal Carcinoma"

_diagnostics, 2025, doi:10.3390/diagnostics15091081_

Round 1
Reviewer 1 Report
Comments and Suggestions for Authors
Thank you for the opportunity to review your manuscript titled “Application of 18F-FDG PET/MR in the Evaluation of Oropharyngeal Carcinoma.” Your work addresses a clinically relevant and timely topic in the field of head and neck oncology, particularly concerning advances in hybrid imaging techniques. However, there are several areas within the manuscript that would benefit from clarification, methodological elaboration, and editorial refinement. These adjustments will help ensure the work meets the standards of scientific rigor and enhances its clarity and reproducibility for the readership. Please find below my detailed comments and suggestions for improvement.
- the introduction needs some improvements so please Consider restructuring the it to follow:(1) Clinical burden (2) Need for accurate staging (3) Limitations of current imaging modalities (4) Rationale for PET/MR (5) Gap in literature (6) Study aims.
End with a concise statement of your hypothesis and purpose. - The comment that “PET/CT has not shown advantages over MR” [5] is oversimplified. Some studies suggest that PET/CT provides superior metabolic characterization, especially in nodal staging. A more balanced discussion would strengthen your argument.
- A full language and grammar review by a native English-speaking editor or professional editing service is strongly recommended before resubmission.
- Authors should clearly separate the two main objectives for better clarity:
(1) comparing PET/MR with CT/MR for staging, and
(2) examining the correlation between imaging biomarkers and biological factors. - authors shall state the study type (e.g., retrospective observational study).
- Authors shall Include in inclusion/exclusion criteria the tumor stage, HPV status.
- Authors shall discuss sample size justification or acknowledge limitations in the Discussion.
- The CT protocol mentions "various models"—this is vague and raises reproducibility concerns. Were all patients scanned with the same equipment and protocol?
- some key acquisition parameters such as scan duration, slice thickness, MR sequences used for soft tissue, DWI details, attenuation correction methods should be mentioned
- it is unclear how many PET/MR findings were correlated with histopathological lymph node positivity. This is crucial for validating PET/MR accuracy.
Comments on the Quality of English Language
A full language and grammar review by a native English-speaking editor or professional editing service is strongly recommended before resubmission.
Author Response
1. the introduction needs some improvements so please Consider restructuring the it to follow:(1) Clinical burden (2) Need for accurate staging (3) Limitations of current imaging modalities (4) Rationale for PET/MR (5) Gap in literature (6) Study aims. End with a concise statement of your hypothesis and purpose.
Response: Thank you for your advice. We have revised the abstract.
2. The comment that “PET/CT has not shown advantages over MR” [5] is oversimplified. Some studies suggest that PET/CT provides superior metabolic characterization, especially in nodal staging. A more balanced discussion would strengthen your argument.
Response: Thank you for your advice. We have revised it.
3. A full language and grammar review by a native English-speaking editor or professional editing service is strongly recommended before resubmission.
Response: Thank you for your advice. We have revised it using language editing provided by MDPI author service.
4. Authors should clearly separate the two main objectives for better clarity:
(1) comparing PET/MR with CT/MR for staging, and
(2) examining the correlation between imaging biomarkers and biological factors.
Response: Thank you for your advice. We have clarified it in the objective of abstractb and introduction.
5. authors shall state the study type (e.g., retrospective observational study).
Response: Thank you for your advice. We have added it in the abstract.
6. Authors shall Include in inclusion/exclusion criteria the tumor stage, HPV status.
Response: Thank you for your advice. We have added it in part 2.1.
7. Authors shall discuss sample size justification or acknowledge limitations in the Discussion.
Response: Thank you for your advice. We have added it in the final paragraph of discussion part.
8. The CT protocol mentions "various models"—this is vague and raises reproducibility concerns. Were all patients scanned with the same equipment and protocol?+
Response: Thank you for your advice. The CT scanner is of the same type. We have revised it in part2.1.
9. some key acquisition parameters such as scan duration, slice thickness, MR sequences used for soft tissue, DWI details, attenuation correction methods should be mentioned
Response: Thank you for your advice. We have added them in part 2.2.
10. it is unclear how many PET/MR findings were correlated with histopathological lymph node positivity. This is crucial for validating PET/MR accuracy.
Response: Thank you for your advice. We have added them in the Supplementary Table 2.
Reviewer 2 Report
Comments and Suggestions for Authors
Dear authors,
even though this is an interesting paper, some issues are present:
- the number of the protocol of the ethics committee is missing;
- it is necessary to underline the fact that the patients should avoid talking during the uptake time of the radiopharmaceutical, since this could impact the evaluation of the primary tumor;
- it has not been described how SUV(max), SUV(mean), SUV(peak), ADC(min), ADC(max), ADC(mean), MTV and TLG were calculated;
- how were PET/MR images qualitatively interpreted before using the aforementioned parameters? Please clarify;
- the reported value of SUVmax of 2.5 for the classification of positive metastatic lymph nodes has an intrinsic limitation due the false positive ratio of immunoreactive nodal activation. This should be stated and specified since the staging classification is based on imaging findings in this paper, as you underlined;
- it is stated that the accuracy of the imaging modalities is based on the surgical specimen as gold standard, however not all the patients performed sthe urgery. Please clarify;
- why volumetric parameters such as MTV and TLG were calculated only by focusing on the primary lesion (line 130) and not also on lymphnodes? Please clarify;
- line 165: the volume of the lesion on PET/MR was calculated on PET or on MR images? Please specify;
- it is necessary to specify the temporal distance between the assessment of tumor markers and PET imaging, their units of measure and the modality used for their quantification;
- similarly to the previous point, how and when p16 was evaluated? Please clarify.
Author Response
Dear authors,
even though this is an interesting paper, some issues are present:
- the number of the protocol of the ethics committee is missing;
Response. Thank you. We have added in part2.1, line104.
- it is necessary to underline the fact that the patients should avoid talking during the uptake time of the radiopharmaceutical, since this could impact the evaluation of the primary tumor;
Response. Thank you for your advice. We have added in part 2.2.
- it has not been described how SUV(max), SUV(mean), SUV(peak), ADC(min), ADC(max), ADC(mean), MTV and TLG were calculated;
Response: Thank you for your advice. We have added and described them in part 2.3.
- how were PET/MR images qualitatively interpreted before using the aforementioned parameters? Please clarify;
Response: Thank you for your advice. We have added and described them in part 2.3.
- the reported value of SUVmax of 2.5 for the classification of positive metastatic lymph nodes has an intrinsic limitation due the false positive ratio of immunoreactive nodal activation. This should be stated and specified since the staging classification is based on imaging findings in this paper, as you underlined;
Response: Thank you for your advice. We have added and described them in part 2.3.
- it is stated that the accuracy of the imaging modalities is based on sthe urgical specimen as gold standard, however not all the patients performed surgery. Please clarify;
Response: Thank you for your advice. We have clarified it in the first sentence of part 3.3.
- why volumetric parameters such as MTV and TLG were calculated only by focusing on the primary lesion (line 130) and not also on lymphnodes? Please clarify;
Response: We have clarified that all parameters were calculated in both primary oropharyngeal lesions and metastatic lymph nodes. Please see line. But the relationship between parameters and Ki67, p16 was only verified in primary lesion because we only detected Ki67 and p16 in primary lesion.
- line 165: the volume of the lesion on PET/MR was calculated on PET or on MR images? Please specify;
Response: We have clarified that the volume of the lesion were seperatly calculated by both CT, MR and PET/MR and then we compared the difference. Please see line 246.
- it is necessary to specify the temporal distance between the assessment of tumor markers and PET imaging, their units of measure and the modality used for their quantification;
Response: Thank you for your advice. We have added the temporal distance between the assessment of tumor markers and PET imaging, please see line 315. The units of tumor markers were added in table 4.
- similarly to the previous point, how and when p16 was evaluated? Please clarify.
Response: Thank you for your advice. We have clarified in part 2.4.
Round 2
Reviewer 1 Report
Comments and Suggestions for Authors
thanks for revising the manuscript
Reviewer 2 Report
Comments and Suggestions for Authors
All the issues have been addressed and the overall quality of the paper has been improved.